



# Instantaneous variance scaling of AIRS profiles using a circular area Monte Carlo approach

Jesse Dorrestijn, Brian H. Kahn, João Teixeira, and Fredrick W. Irion

Jet Propulsion Laboratory, California Institute of Technology, 4800 Oak Grove Drive, Pasadena, CA 91109, USA

*Correspondence to:* Jesse Dorrestijn (jesse.dorrestijn@jpl.nasa.gov)

**Abstract.** Satellite observations are used to study the variance scaling of temperature and water vapor in the atmosphere. A high resolution of 13.5 km at nadir, instead of 45 km as in previous Atmospheric Infrared Sounder (AIRS) studies, enables the derivation of the variance-scaling exponents down to length scales of ∼55 km. With the variable-size circular area Monte Carlo approach the exponents can be computed instantaneously along the track of Aqua, which gives more insight into the scaling behavior of the atmospheric variables in individual Level 2 satellite granules. Scaling exponents are shown to fluctuate heavily between $\beta = -1$ and $\beta = -3$ at the larger scales, while at the smaller scales they are often closer to $\beta = -2$, and they decrease a bit for moisture at the smallest scales that are considered. Outside the tropics, the temperature large-scale variance-scaling exponent is often close to -3 due to large temperature slopes that are present along the track of Aqua, likely as a result of geostrophic turbulence. Around the tropics, this exponent is often closer to -1, because the tropical atmosphere is dominated by smaller-scale processes such as moist convection, leading to an observable reverse scale break. In contrast, water vapor is shown to have large-scale exponents often close to -3 around the tropics, because there, large-scale water vapor slopes are common along the Aqua track. Furthermore, the scale-break length scale turns out to be highly variable and shows a large spread. The presented variance-scaling results are of importance for cloud parameterization purposes.

## 1  Introduction

The atmosphere distributes its kinetic and thermal energy over its entire range of scales. Energy that is present at larger scales tends to cascade towards the smaller scales where kinetic energy is turned into heat by dissipation at the Kolmogorov length scale (Hunt and Vassilicos, 1991). A complicating factor is that Earth's atmosphere as a whole is near two-dimensional, while at the smaller scales it is three-dimensional. In two-dimensional turbulence, enstrophy is conserved (Kraichnan, 1967; Leith, 1968) such that energy that is injected at the smaller scales can also be transferred towards the larger scales (Lindborg, 1999; Charney, 1971; Fjørtoft, 1953). Furthermore, numerous processes affect the atmosphere at different length scales (e.g. the large-scale planetary circulation, synoptic systems, clouds, deep convection, shallow convection, turbulence, and molecular





diffusion). Therefore, the rate at which the variance of atmospheric variables changes as a function of length scale, the *variance scaling*, is not uniform over the entire range of scales.

Observations have been used to demonstrate that atmospheric variables satisfy specific scaling laws. Julian et al. (1970) showed that at the larger scales ($> 1,500$ km) the power spectra tend to be close to a $k^{-3}$ law. At smaller scales ($< 500 - 700$

km) the slope of the spectra is shallower and prefers a $k^{-5/3}$ law, which transition has been clearly demonstrated with aircraft observations of wind and temperature by Nastrom and Gage (1985). Their variance power spectra diagram (Fig. 3 of Nastrom and Gage (1985)) has often been reproduced (e.g., Lindborg, 1999; Tung and Orlando, 2003; Palmer, 2012).

The precise variance scaling of these variables is however, complicated. Kahn and Teixeira (2009) (KT09 hereafter) have used satellite observations of temperature and water vapor to derive sensitivities to several factors such as the location on

Earth, the season, the presence of clouds and surface conditions (land or ocean). The underlying cause of these sensitivities and even more complicated phenomena such as scale breaks and reverse scale breaks (demonstrated to exist by KT09), are not yet fully understood. One of the reasons is that it is hard to obtain extensive observational data sets corresponding to well-defined atmospheric conditions for a large range of scales. Furthermore, clear sensitivities of scaling in an area appear only after averaging over time periods on the order of a season (KT09).

A myriad of studies of atmospheric variability with numerical models have been performed, e.g., Jonker et al. (1999) used a large-eddy simulation (LES) model to show that passive scalars in a turbulent field can possess different power spectra than the thermodynamic variables themselves. Cusack et al. (1999) studied horizontal variance of moisture with global weather model analyses data and constructed a cloud parameterization. Hamilton et al. (2008) showed the transition from a steep $k^{-3}$ law to a shallower $k^{-5/3}$ law in the kinetic energy spectrum of a general circulation model (GCM).

Just as with observations, simulations have their restrictions considering the range of scales that they cover: due to computational restrictions LES models can not yet accurately simulate synoptic systems, while on the other hand GCMs and cloud resolving models (CRMs) do not resolve smaller-scale processes (e.g. turbulence, shallow convection) that affect the variance scaling exponents. Parameterizations of these unresolved processes are often based on assumptions about variance scaling (Bogenschutz and Krueger, 2013; Tompkins, 2002; Teixeira and Hogan, 2002; Larson et al., 2002), and therefore, can not be

used to derive variance scaling exponents at or close to the subgrid-scales. In short, there is a need for numerical and observational studies that deliver statistics of the scaling exponents at an increased range of length scales, in particular near the GCM subgrid-scale (Kahn et al., 2011).

To follow up on the work described by KT09, this paper presents a new variance-scaling method which is applied to satellite data derived atmospheric temperature and water vapor mass mixing ratio with a higher horizontal resolution (three times

higher in both horizontal directions). The new variance scaling method enables instantaneous variance scaling along the track of Earth observing satellites. For a given horizontal two-dimensional atmospheric field (e.g. temperature, water vapor) at a certain altitude in the atmosphere, standard deviations are calculated over areas for a range of length scales from which variance scaling exponents are derived. Areas are chosen to be of *circular* shape and can be placed along the track of a satellite. Variance scaling exponents are estimated by varying the diameter of the circular areas. To get adequate estimates, a Monte

Carlo method is employed, which uses randomly placed smaller circles inside a large circle.



The paper is organized as follows: Section 2 describes the atmospheric datasets, which is followed by a section that introduces the new variance-scaling method (Section 3). Variance-scaling results are presented in Section 4. Finally, Section 5 discusses the implications of the findings, draws conclusions and presents future work that is possible with this newly developed approach.

## 2   Datasets

Temperature and water vapor mass mixing ratio profiles are derived from observations made by the Atmospheric Infrared Sounder (AIRS) onboard the Aqua spacecraft (Aumann et al., 2003; Parkinson, 2003; Chahine et al., 2006). The Aqua spacecraft is part of the Earth Observing System (EOS), an international program centered at the National Aeronautics and Space Administration (NASA), and shares its orbit with the other satellites that form the afternoon satellite constellation A-Train (Vicente et al., 2006; Stephens et al., 2008). It orbits the Earth at a ∼705 km altitude in a sun-synchronous near-polar orbit, which means that it orbits from pole to pole crossing the equator at fixed times, i.e. around 1:30 p.m. local time on *ascending* (northwestwards) tracks and 1:30 a.m. on *descending* (southwestwards) tracks, which enables the instruments to have a near-global coverage each day.

### 2.1   AIRS-AMSU-HSB

AIRS is a cross-track-scanning spectrometer with 2,378 infrared (IR) channels that cover a spectral range from 3.7 to 15.4 $\mu$m (Aumann et al., 2003). There are 90 AIRS-IR ground footprints per swath of ∼1,650 km (depicted in Fig. 1 of Aumann et al. (2003)), which results in a horizontal resolution of 13.5 km at nadir. Additionally, the instrument has four visible and near-infrared channels with ranges between 0.40 and 0.94 $\mu$m with a horizontal resolution of 2.28 km at nadir (Gautier et al., 2003). The self-calibrating instrument enables the estimation of vertical profiles of several atmospheric variables (e.g., temperature, humidity) and minor gases (e.g., ozone, carbon dioxide) from the surface up to an altitude of 40 km with a quality approaching conventional radiosonde soundings and a vertical resolution of one kilometer (Chahine et al., 2006).

AIRS is accompanied by two synchronized and aligned microwave instruments: the Advanced Microwave Sounding Unit (AMSU), a two-unit microwave radiometer with 15 channels that are sensitive to frequencies between 23 and 89 GHz, around the 60-GHz oxygen band, with a horizontal resolution of 45 km at nadir and the Humidity Sounder for Brazil (HSB), a four-channel radiometer sensitive to frequencies between 150 and 190 GHz, concentrated around the 183-GHz water vapor line, with a horizontal resolution of 13.5 km at nadir (Lambrigtsen and Calheiros, 2003).

The microwave instruments are used to detect clouds and appropriately correct the IR spectra by applying a process called *cloud clearing* (Susskind et al., 2003). During the process, the horizontal resolution is reduced from 13.5 km to 45 km, because to obtain cloud-cleared spectra, each AMSU footprint is combined with nine AIRS footprints. The cloud-cleared spectra are used to derive temperature and humidity profiles in the AIRS-AMSU and AIRS-AMSU-HSB data products (Chahine et al., 2006). A third product, AIRS-IR (also called AIRS-only) is a control product that does not use microwave instruments, but has the same horizontal resolution as the AIRS-AMSU and AIRS-AMSU-HSB data products.




The three-instrument AIRS-suite enables the estimation of three-dimensional atmospheric fields along the orbit of Aqua, since 30 Aug. 2002 to present (until 5 Feb 2003 for HSB). Swath measurements are collected every six minutes in *granules*, which are processed into Level 2 data products. Every day there are 240 granules available, each consisting of 30 times 45 fields or vertical profiles depending on the variable under consideration. Fig. 1a displays an AIRS-AMSU-HSB Version 6 (v6) temperature field at 500 hPa in the first granule that is available at NASA's Goddard Earth Sciences (GES) Data and Information Services Center (DISC). Details about the AIRSv6 datasets can be found in Susskind et al. (2014).

## 2.2 AIRS-OE

Methods are being developed to deal with clouds during the retrieval process without reducing the horizontal resolution of the temperature and humidity fields that are retrieved with an infrared sounder. The Optimal Estimation Retrieval System for AIRS (AIRS-OE), introduced by Irion et al. (2017), is such a method and its temperature and humidity derived fields are used here in addition to the three coarser-resolution AIRS data products described in the previous subsection. In their method, which is based on retrieval methods of Bowman et al. (2006) and Rodgers (2000), cloud detection and cloud property estimation have been performed using data from the Moderate Resolution Imaging Spectroradiometer (MODIS) instrument, a 36-channel cross-track scanning radiometer measuring visible and infrared radiation in a spectral range from 0.415 to 14.235 $\mu$m with a horizontal resolution of 250-1,000 m at nadir and a swath width of 2,330 km, also onboard the EOS Aqua spacecraft (King et al., 2003; Parkinson, 2003; Platnick et al., 2003).

AIRS-OE data products are created without using microwave instruments, but use single-footprint (instead of nine) IR observations to derive vertical profiles of temperature and water vapor. As a result, AIRS-OE provides vertical profiles of temperature and water vapor with a 3x3 times higher horizontal resolution than the resolution of the AIRS-AMSU-HSBv6 fields. Fig. 1b depicts an AIRS-OE temperature field in the first granule as well. It contains up to 90 retrievals in a swath and up to 135 retrievals along the track. It can be seen that AIRS-OE captures small-scale temperature variability that is not captured with AIRS-AMSU-HSBv6.

The advantage of using AIRS-OE over the AIRS-AMSU(-HSB)v6 instrument suite retrievals is the higher resolution, on the other hand there is the disadvantage that below IR-blocking clouds there are no retrievals. Moisture fields of AIRS-AMSU-HSBv6 and AIRS-OE are displayed in Fig. 1c-d: apart from the missing data in AIRS-OE the fields are quite similar. For further analysis and comparison of these two data products, the reader is referred to Irion et al. (2017).

To summarize, in this paper, four AIRS data products are considered: three products with a horizontal resolution of 45 km : AIRS-IRv6, AIRS-AMSUv6 and AIRS-AMSU-HSBv6 and the 13.5 km horizontal resolution product AIRS-OE.

## 3 Method

The method that is presented in this paper is first of all similar to the "poor man's spectral analysis" method described by Lorenz (1979) and employed as well by KT09: instead of calculating power spectra and measuring slopes $\beta$ within the diagrams,





standard deviations are used to calculate scaling exponents corresponding to a range of length scales. In this paper, the scaling exponents obtained using standard deviations are referred to as "variance scaling" exponents.

If a power-law relation exists between the standard deviation and the length scale, then given two length scales $l_1 < l_2$ with standard deviations $\sigma_1$ and $\sigma_2$, the scaling exponent $\alpha$ is:

$$\alpha = \frac{\log_e(\sigma_2) - \log_e(\sigma_1)}{\log_e(l_2) - \log_e(l_1)}.$$

When plotting the standard deviation as a function of length scale, while using logarithmically scaled horizontal and vertical axes, the scaling exponent $\alpha$ determines the slope of the line from $(l_1, \sigma_1)$ to $(l_2, \sigma_2)$. This line is straight if a power-law relation exists and is half as steep as for variances, which can equivalently be used instead of standard deviations to calculate the variance scaling exponents (Vogelzang et al., 2015).

Analogously to KT09, in this paper $\alpha_L$ is defined as the "large-scale" scaling exponent for length scales between $6°$ and $12°$, $\alpha_S$ for "small" length scales between $1.5°$ and $4°$. In addition, $\alpha_T$ is defined as the scaling exponent for "tiny" length scales between $0.5°$ and $1.5°$. Length scales are expressed in degrees over great circles and are assumed to not change with latitude or longitude. To connect the computed $\alpha$ values to the power spectra line slopes $\beta$, $\alpha$ values are converted in $\beta$ values by using the equation on page 5563 of KT09: $\beta = -(2\alpha + 1)$ (see also Davis et al. (1996); Yu et al. (2017)). The well-known $\beta = -5/3$ and
$\beta = -3$ correspond to $\alpha = 1/3$ and $\alpha = 1$, respectively. The next subsection gives a description of the estimation of standard deviations along the track of Aqua.

### 3.1 Circular approach

Standard deviations are computed over *circular areas* with diameter $l$. The largest length scale is determined by the swath width of AIRS, i.e. $L = 15.4°$. In that case, a circle with radius $7.7°$ is positioned with its center at Aqua's track (at nadir) after which
the standard deviation of the temperature and water vapor values that are inside it are calculated. A depiction of a water vapor field at 500 hPa using AIRS-OE-retrievals that are inside a circle with $15.4°$ diameter is presented in Fig. 2a. The smallest length scale is determined by the horizontal resolution of the observations. Here, the minimum number of retrievals that are required to get a (contributing) standard deviation from a circle is five, as in KT09. Taking this requirement in consideration, for AIRS-OE the smallest length scale, and hence the smallest diameter of the circles, is chosen to be $l = 0.5°$ and for the three
coarser-resolution AIRSv6 data products this is $l = 1.5°$.

### 3.2 Monte-Carlo method

To obtain the standard deviations corresponding to length scales smaller than $L = 15.4°$, i.e. length scales $0.5° \leq l < 15.4°$, smaller circles are randomly placed inside the largest circle with diameter $15.4°$. So, the radius of the circle is reduced to obtain standard deviations corresponding to smaller length scales $l$. Given a smaller circle with diameter $l < 15.4°$, it is placed at a
random location inside the largest circle, and such that it is entirely inside it, and the standard deviation of the temperature or water vapor values that are located in the smaller circle are computed.



In order to get a better estimate, a *Monte-Carlo* estimation procedure is employed: the random placement is repeated 10,000 times, which means that 10,000 circles are placed randomly inside the largest circle, and their average standard deviation is used as an estimate of the standard deviation corresponding to the length scale $l$. A subtle note on random placement: this should be done such that the 10,000 smaller circles cover the largest circle uniformly. This procedure is repeated for all length scales $0.5° \leq l < 15.4°$ (down to $1.5°$ when using AIRSv6 products). Two of these 10,000 smaller ($l = 6°$) circles that are entirely contained in the $15.4°$-diameter circle are displayed in Fig. 2b.

### 3.3   Instantaneous variance scaling along the track

A special feature of the method is that the $15.4°$-diameter circle can move *along with the orbit* of the Aqua satellite. To be precise, its center can move along with the track of the orbit's nadir to estimate standard deviations and hence the variance scaling of a horizontal circular area around the track. This works fine as long as circles are entirely contained in a single granule. If the $15.4°$-diameter circle is close to the edge and partly covers two granules, the two granules can be glued together. By repeating this procedure of glueing granules, the variance scaling can be derived *instantaneously* (i.e. without time averaging) along the track of the Aqua satellite and for as many granules as desired.

By varying the diameter of the circular areas, while covering the selected area in a uniform way, standard deviations can be estimated in the area under consideration. The diameter of the circles can vary with arbitrary small increments, and is chosen to vary with increments of $0.5°$, which gives a sufficient resolution in the variance-scaling plots that are introduced in Section 4. After the calculations of the standard deviations as a function of length scale, the three $\alpha$ exponents (i.e. the slopes) are estimated by a least squares fit (Weisstein, 2017). Furthermore, $\beta_L$, $\beta_S$ and $\beta_T$ are the analogs of $\alpha_L$, $\alpha_S$ and $\alpha_T$ and will sometimes be used instead of the $\alpha$ values, because the $\beta$ values, such as $\beta = -3$ and $\beta = -5/3$, are more commonly used in the literature.

### 3.4   Scale-break detection

To examine the length scale at which the variance scaling exponents change, for example from a $\beta = -3$ to a $\beta = -5/3$ slope, the standard deviation as a function of length scale is approximated by two power laws, which is equivalent to fitting two straight lines optimally in a double-log scaled figure. When using the AIRS-OE data product, which covers a larger range of scales, the double scale break is examined by fitting three straight lines in the variance scaling plots. To do this optimally, all possible (double) scale break positions $l \in \{1°, 1.5°, \dots, 15°\}$ are calculated to find the two (three) straight lines that minimize the error with the data.



## 4 Results

### 4.1 Variance-scaling diagrams

The aim is to construct variance-scaling diagrams that are similar to Fig. 3 of Nastrom and Gage (1985). First, the position of the $15.4°$-diameter circular area is fixed on the track of the satellite and standard deviations of temperature or water vapor

at a certain pressure level are calculated for each of the length scales $0.5° \leq l \leq 15.4°$ (as described in Section 3). Then, these standard deviations are plotted on the logarithmically scaled y-axis as a function of the length scale $l$, which is on the logarithmically scaled x-axis.

Fig. 3 displays the standard deviations of the temperatures at 500 hPa at four locations on the track of Aqua, calculated with circular areas within four $15.4°$-diameter circles having their centers on the track. The four locations are chosen because their

diagrams give an overview of typical behavior of the scaling, but note that the precise scaling patterns can change drastically at these locations depending on the atmospheric circumstances. The four AIRS data products are included to get an impression of the uncertainty of these scaling diagrams due to sampling differences.

The standard deviation usually increases as a function of length scale, only in Fig. 3c the standard deviation decreases at the larger length scales when AIRS-OE is used. In Fig. 3a, this increase is not constant: at the larger scales, it is close to $\beta = -3$

and at the smaller scales $\beta$ increases up to $\beta = -5/3$ for AIRS-OE. In this panel, the slope changes between the length scales $l = 9°$ and $l = 11°$, which is an example of a scale break. Observe that, in Fig. 3c, the slope at the smaller scales is steeper than at the larger scales, which is an example of a reverse scale break which has been reported before by KT09 for specific humidity. In Fig. 3d, there is no clear scale break at all.

The differences of the standard deviations between the three coarser-resolution AIRS data products are small and differences

that arise can partly be attributed to blocking clouds. AIRS-OE tends to give higher standard deviations, most notably in Fig. 3b, which is to be expected because of its higher spatial resolution. Further discrepancies can partly be attributed to gaps in the temperature or water vapor fields due to unsuccessful retrievals (e.g. of low quality) mostly in AIRS-OE. Observe that the relative differences between the slopes of the four lines, which are of main interest, are smaller than the relative differences in the actual values.

The corresponding moisture plots are given in Fig. 4. The scaling diagram in Fig. 4a is similar to the temperature diagram in Fig. 3a, but the slopes are even closer to $\beta = -5/3$ at the smaller length scales. In Fig. 4b, the reverse scale break is clearly visible around $l = 9°$. In Fig. 4c, the discrepancies between the four data products are significant at the larger scales, where AIRS-OE displays a decreasing standard deviation as a function of length scale. At the smaller scales the slopes are similar and between $\beta = -3$ and $\beta = -5/3$. In Fig. 4d, all slopes are close to $\beta = -3$ at the larger scales, then slowly increase to $\beta = -1$

(i.e. $\alpha = 0$) around $l = 7°$ and decrease again at smaller length scales.

### 4.2 Variance scaling along the track

Instead of plotting the entire variance scaling diagram as has been done in the previous subsection, here the focus is entirely on the scaling exponents $\alpha_L$, $\alpha_S$ and $\alpha_T$, along the track of Aqua. The center of the $15.4°$-diameter circle is positioned at Aqua's



nadir at the first swath of the second granule that is available in the database and is moved 84 min Aqua space flight time along the track, corresponding to 14 granules, while estimating scaling exponents of the temperature and water vapor at 500 hPa. Note that this 84 min dataset is a small subset of the entire multi-year ($\sim$ 15 year) AIRS dataset. The centers of consecutive $15.4°$-diameter circles are chosen to be 8 seconds apart from each other, corresponding to the time it takes to make one AMSU

swath (i.e. 6 min divided by 45 swaths). Due to overlap there are strong correlations between the exponents calculated on consecutive circular areas along the track.

Fig. 5a displays the three scaling exponents derived from AIRS-OE temperature retrievals. Observe that the exponent $\alpha_L$ fluctuates between $\alpha = 0$ and $\alpha = 1$ (left reversed vertical axis) which corresponds to $\beta$ values between $\beta = -1$ and $\beta = -3$ (right vertical axis). The exponent $\alpha_S$ fluctuates around $\alpha = 1/3$ corresponding to $\beta = -5/3$ and has a smaller range than

$\alpha_L$. The exponent $\alpha_T$ has a slightly smaller range than $\alpha_S$ and stays usually between $\alpha = 1/2$ and $\alpha = 1/3$ corresponding to $\beta = -2$ and $\beta = -5/3$.

The standard deviation estimates from which these scaling exponents are calculated are shown in Fig. 5b. The lowest line is the standard deviation corresponding to $l = 0.5°$, the line above it is the standard deviation corresponding to $l = 1.0°$. It is clear that standard deviations are, usually, increasing as a function of length scale.

Notable is that the local maximum values of the standard deviation of $15.4°$ tend to co-align with the local maximum values of $\alpha_L$, corresponding to the local minimum values near $\beta = -3$. A large standard deviation at the larger scales indicates that there is a large-scale slope in temperature along the satellite track, which correlates strongly with $\alpha_L$ as will be shown later. The location on Earth can be derived from Fig. 5c which displays the longitude and latitude of Aqua's nadir, which is the center of the $15.4°$-diameter circle.

A large separation of the three lines in Fig. 5a, indicates a large difference in slopes and hence the existence of scale breaks. For example at time 75 min, the large-scale exponent is close to $\alpha = 1$ ($\beta = -3$), while $\alpha_S$ and $\alpha_T$ are smaller, such that $\beta$ increases at smaller length scales. Close to the equator around 33 min, the roles are reversed: the large-scale exponent is close to zero, the tiny scale exponent close to $\alpha = 1/2$ and the small-scale exponent in between these two, indicating the existence of a double reverse scale break. Just before 60 min, the three values are almost equal such that that there are no clear scale breaks.

Its variance-scaling plot will be similar to Fig. 3d for which the three values are almost equal as well (around 46 min).

The corresponding moisture scaling exponents are shown in Fig. 6a. The moisture $\alpha_L$ is in the same range as the temperature large-scale exponent. The moisture small-scale exponent $\alpha_S$ has slightly larger values than the temperature equivalent. The tiny-scale exponent $\alpha_T$ is significantly higher for moisture than for temperature. Fig. 6b shows the standard deviations of the water vapor mass mixing ratio. It is clearly visible that in the tropics (the location on Earth can again be derived from Fig. 5c)

the standard deviation of the water vapor is larger than outside the tropics; compare e.g. granule 231 with granule 235. Local maximum values are again to some extent co-aligned with the local maximum values of $\alpha_L$.

### 4.3 Variance scaling at several pressure levels

KT09 showed that the scaling exponents are (among other factors) sensitive to the altitude in the atmosphere, the surface type and the presence of clouds. Therefore, variance-scaling exponents are calculated along the same track segment at three pressure





levels (300 hPa, 500 hPa and 850 hPa) and displayed in Fig. 7 for temperature and in Fig. 8 for moisture. The results are noisier at the lower pressure level (compare Fig. 7c to Fig. 7a,b) because there, the number of successful retrievals, *the yield*, is lower.

The three coarser-resolution AIRS products give comparable results when the yield is high, therefore of these three products only AIRS-AMSU derived exponents are shown in Fig. 7 and Fig. 8. In Fig. 7, it can be seen that the large-scale exponent $\alpha_L$

(blue dash-dotted line) for temperature fluctuates roughly between $\alpha = 0$ and $\alpha = 1$, apart from a part at the lower 850 hPa pressure level (Fig. 7c) around the South Pole (granule 235) where the yield happens to be exceptionally low. The small-scale exponent $\alpha_S$ (red solid line) fluctuates in a smaller range from $\alpha = 0.25$ up to $\alpha = 0.75$, apart from the interval around 24 min in which it almost reaches $\alpha = 1$ at the lowest level 850 hPa.

The reverse scale break around the tropics (granule 231) is clearly visible at 500 hPa and to a lesser extent at the other levels,

which is in agreement with the results of KT09. The scaling exponent $\alpha_T$ (cyan dashed line) that is derived from the AIRS-OE product fluctuates between $\alpha = 0.25$ and $\alpha = 0.5$ most of the time at all three levels. Sensitivities to the surface type and cloud fraction (not shown in the figures) are less obvious and become clear only after averaging over long time series, which has been performed by KT09.

Observe that in Fig. 8, the moisture scaling exponents fluctuate faster than the temperature scaling exponents at all levels.

The large-scale exponents $\alpha_L$ fluctuates again between $\alpha = 0$ and $\alpha = 1$. The small-scale exponent $\alpha_S$ is in the same range as was the case for temperature, but the tiny-scale exponent $\alpha_T$ is more often larger than $\alpha = 0.5$ (below $\beta = -2$) and has a larger range for moisture than for temperature (compare Fig. 8 and Fig. 7). In granule 233 at 300 hPa, $\alpha_T$ even becomes much larger than $\alpha = 1$, which means that the slope in the variance-scaling figure is steeper than $\beta = -3$.

### 4.4   Variance-scaling exponent statistics

Statistics of the scaling exponents of the temperature and water vapor at 500 hPa are shown in Fig. 9. The probability density functions (PDFs) of the exponents are derived from five days of observations. To increase the speed of the calculations only 100 circles are used in the Monte-Carlo estimation method, leading to slightly too large extreme values of the exponents. It can be seen that the large-scale exponents have a larger range than the small-scale exponents, both for temperature and water vapor. Furthermore, the small-scale exponents have more symmetric distributions and the PDFs attain their maximum values

close to $\alpha = 0.5$.

The asymmetry of the large-scale exponent distribution for temperature is caused by the different values of the exponent in and outside the tropics. In the tropics this exponent tends to be close to $\alpha = 0$ and outside it is often closer to $\alpha = 1$. The small-scale exponent does not have such a strong latitude dependency and therefore does not display such a skewed distribution. The large-scale exponent for water vapor is skewed in the other direction than the temperature large-scale exponent, because

$\alpha$ tends to be close to 1 in the tropics and outside often closer to 0.

These kind of statistics are very valuable to improve cloud parameterizations that make use of PDF schemes. Especially the distribution of $\alpha_T$, which is not shown, because the AIRS-OE data product is still in preparation and the 16 granules form too small a dataset to make reliable statistics. The statistics that are shown, are intended to be an example of what is possible with the new scaling approach combined with AIRS datasets. Producing reliable statistics with the AIRS dataset, for example





derived from 15 years of observations and using the method with 10,000 circles instead of 100, lies out of the scope of this paper that mainly introduces the new circular variance scaling approach.

## 4.5 Correlation analysis

To relate the variance-scaling exponents to physical quantities, a correlation analysis is performed using AIRS-AMSU. The

results are shown in Fig. 10. The 686 values (corresponding to the same number of swaths) cover a slightly larger part of the track than the 84 min segment that was used in subsections 4.2 and 4.3. The strongest correlation is found between the large-scale exponent $\beta_L$ and the absolute value of the mean temperature change (the slope) in the direction of the track of Aqua at nadir (Fig. 10a). The temperature change that is considered is the difference between the average temperature at 500 hPa measured over consecutive 15.4°-diameter circles (which centers are 8 seconds apart from each other). If there is a large

temperature slope, then the exponent tends to be close to $\beta = -3$, corresponding to geostrophic turbulence or a transition from land to ocean. If there is no slope, the exponent is closer to $\beta = -1$, corresponding to an atmosphere dominated by smaller-scale processes, such as tropical convection.

The large-scale exponent is also well-correlated with the standard deviation in the 15.4°-diameter circle (Fig. 10b), which confirms the alignment of peaks observed in Fig. 5a,b. For moisture, strong correlations (above 0.50) are found between $\alpha_L$

and the standard deviation in the 15.4°-diameter circle (Fig. 10d) and the absolute value of the moisture slope along the track (Fig. 10e). The outgoing longwave radiation (OLR) (Fig. 10g) does not display a strong correlation with water vapor and even less with the temperature (Fig. 10i). The exponents $\beta_L$ and $\beta_S$ are positively correlated for both temperature (Fig. 10c) and water vapor (Fig. 10f). The surface type (land or ocean) is not strongly correlated with the temperature or moisture large-scale scaling exponents at the lower pressure level 850 hPa (not shown). Finally, the cloud fraction does not show strong correlations

with the scaling exponents at 500 hPa (Fig. 10h).

## 4.6 Scale-break-detection results

Fig. 11 presents scale-break-detection results. From these two plots it is already clear that the scale-break length scale varies significantly along the track of Aqua: the length scale of the scale break in the top panel is two times larger than in the bottom panel (9° and 4.5°, respectively). The scale-break length scale fluctuates heavily between 2° and 15° along the track of Aqua

at the three pressure levels, for all AIRS data products and both for temperature and water vapor (not shown in figure).

To get more insight in the distribution of the scale-break length scale, probability density functions (PDFs) are shown in Fig. 12. To produce these PDFs the range of AIRS-OE is set to $1.5° \leq l \leq 15.4°$, such that it has the same range as the three other AIRS data products. The maximum of the PDFs is attained between 7° (i.e. ~770 km) and 10.5° for temperature and between 5° and 8° for water vapor at 850 hPa (Fig. 12e,f), between 8° and 11° for temperature at 500 hPa (Fig. 12c) and around 9° for

water vapor at 500 hPa (Fig. 12d), between 5.5° and 9° for temperature at 300 hPa (Fig. 12a) and around 9° for water vapor at 300 hPa (Fig. 12b), which is to some extent in agreement with 500-700 km and 450-750 km scale break length scales reported by Gage and Nastrom (1985) and Tung and Orlando (2003), respectively. The PDFs of temperature (Fig. 12a,c) have a less well-defined maximum than water vapor (Fig. 12b,d,f), apart from temperature at 850 hPa (Fig. 12e), in which a clear peak is



visible around $7°$. In the next section we provide a tentative physical explanation for the large spread of the scale-break length scale.

Finally, a double scale break detection is applied to the AIRS-OE dataset for its full range, i.e. $0.5° \leq l \leq 15.4°$. Temperature and water vapor examples are shown in Fig. 13. This again shows the variety of variance scaling that can be observed in

the atmosphere. Observe that the slope gets steeper at the smaller scales (below $1.5°$). This can be of importance for cloud parameterizations that are based on PDF schemes. Less variance is present at the smaller scales than is expected from a simple extrapolation of the exponents at the larger scales to the smaller scales; this has been recognized as well by KT09 and Kahn et al. (2011).

## 5   Discussion and conclusions

We show that individual granules of satellite Level 2 data can quantify the variability of the atmosphere. In particular, the variance scaling exponents of temperature and water vapor mass mixing ratio have been derived using data stemming from the AIRS instrument suite onboard the Aqua satellite. These exponents are frequently close to the $\beta = -5/3$ and $\beta = -3$ values, however, deviations from these values are very common. The large-scale exponents $\beta_L$, corresponding to the length scales $6°$ to $12°$, fluctuate between -3 and -1.

The precise value of the large-scale exponent depends strongly on the standard deviation of temperature and water vapor content on larger areas. The temperature and moisture slope along the track also affects the large-scale exponent. When the large-scale fluctuations are strong, then the large-scale exponent is close to -3, which is the preferred value in case there is a large slope in the temperature (water vapor) field along the track. Such a temperature slope can be expected off the equator when dynamics are ruled by geostrophic turbulence (Charney, 1971). When the large-scale temperature fluctuations are small,

then smaller-scale fluctuations easily decrease this value to -1. This happens often in the tropics (e.g. Fig.7b, granule 231), where small-scale fluctuations that are likely the result of deep convection, can be stronger than the large-scale fluctuations. In contrast, for moisture, such large-scale slopes can be expected close to and in the tropics, because there moisture differences are large: we have indeed seen in Fig. 6 that $\beta = -3$ is not so common off the tropics for the large-scale water vapor exponent $\beta_L$.

The small-scale variance scaling exponents $\beta_S$, corresponding to length scales from $1.5°$ to $4°$ are more often close to $\beta = -2$, and less often close to $\beta = -3$. By using single-footprint AIRS data, we show that at the smaller scales from $0.5°$ to $1.5°$, the exponents $\beta_T$ are closer to $\beta = -2$ for temperature and slightly lower (between -2 and -3) for moisture. The PDFs of the small-scale exponents $\alpha_S$ (Fig. 9) show a maximum around $\alpha = 0.5$ for both temperature and water vapor, which is perhaps a surprise if one expects these exponents to have a preferred value of close to $\alpha = 1/3$, i.e. $\beta = -5/3$. Variance-

scaling exponents that are close to $\beta = -2$, i.e. $\alpha = 0.5$, indicate that the variance of the temperature is proportional to the length scale $l$. The preferred slope of $\beta = -2$ can, however, not be clearly explained by the authors.

A special feature of the method is the usage of circles to calculate the standard deviations. It is an interesting question what the shape of an area should be if one aims to estimate the variance scaling in the atmosphere at a certain pressure level.





Rectangles have been used before, e.g. by KT09, and have been accepted by the community to be a correct shape to compute variance scaling exponents, because horizontal sections of GCM columns, for which variance-scaling exponents can be useful in cloud parameterizations, are often (nearly) rectangular.

The orientation of a rectangle or square should not be of major importance when calculating variance scaling exponents. Therefore, one could argue that one could use a slight rotation of the rectangle to calculate the variance scaling exponents and take the average of the values of the rotated and non-rotated rectangles. Then, the rectangle can be turned slightly again and again, while calculating scaling exponents. If one continues turning the rectangle, and makes the turning angles infinitesimally small, the resulting shape is a circle. This procedure of rotating can be performed with any arbitrary shape, with the circle as a final result, and therefore, the circle is the "optimal shape" to calculate variance scaling exponents. It is optimal in case

rotational symmetry is desired, for example, because the underlying field is isotropic. This is in line with Pressel and Collins (2012) who found that water vapor variance scaling is approximately isotropic.

A major advantage of using a "poor man's spectral analysis" method is that relatively small datasets are sufficient to estimate variance scaling exponents. Reliable spectral power diagrams of observational data arise only after averaging over relatively large datasets. For example, Nastrom and Gage (1985) obtained their spectral power diagrams by averaging over observations

collected during 6,000 commercial aircraft flights. Furthermore, spatial variances can be calculated in the case of missing data in which case conventional spectral analysis can not be employed (Vogelzang et al., 2015).

In this study, variance-scaling exponents have been computed instantaneously, without using multiple satellite overpasses. The exponents are derived from satellite observations that are made during at most a few minutes, so the scaling method is strictly speaking not entirely instantaneous. However, compared to the months of measurements used by KT09 and Pressel and

Collins (2012) to compute exponents, our method uses relatively small datasets, which justifies the term instantaneous.

The scale-break-detection results have shown that there is a preference for the scale-break length scale between $7°$ (at 850 hPa) and $9°$ (at 500 hPa and 300 hPa). This is slightly larger than the 500-700 km and 450-750 km scale-brake length scales reported by Gage and Nastrom (1985) and Tung and Orlando (2003), respectively, and smaller than the 1,000 km reported by Bacmeister et al. (1996). The spread around this value is large and this preference is only visible in the distribution of the

scale-break length scale, i.e. where the probability density function attains its maximum. An explanation for the large spread is that convective systems exists of all different sizes, thereby increasing the initially present slope of $\beta = -3$ to higher values at different length scales. A larger convective system will give a larger value of the scale-break length scale. In case no large-scale slope is present, the reverse scale-break length scale depends also heavily on the size of the convective systems: the larger the size, the larger the scale-break length scale. Furthermore, since we measure scale-break length scales along the track, when the

satellite moves from a regime where there is a scale-break at a small scale (say $2°$) to a regime where there is a scale-break at a large scale (say $15°$), then all intermediate scale-break length-scales will be attained during the overpass from the first to the second regime. This explains the continuity of the scale-break length scale PDFs.

The variance-scaling exponent $\beta_T$ for moisture has been shown to attain smaller values, i.e. closer to $\beta = -3$, than the exponent $\beta_S$, which means that less variance is present at the length scales between $0.5°$ and $1.5°$ than at length scales between

$1.5°$ and $4°$, which has also been reported in Kahn et al. (2011). This can be of significance for parameterizations of clouds in




GCMs that use assumptions about the variability of moisture content at length scales below the grid scale, for example when a PDF method is used in the cloud parameterizations (Tompkins, 2002; Teixeira and Hogan, 2002; Teixeira and Reynolds, 2008).

Future work that is possible with the new variance scaling method presented here includes detailed examination of the variance scaling of cyclones at different stages of their life cycles, as has been performed with numerical simulations by Waite

5 and Snyder (2013). This is possible because the method enables instantaneous derivation of the variance scaling diagrams. After locating the cyclones in satellite data and tracking them for a few days, the variance scaling diagrams can be computed for each stage of their development.

*Data availability.* The AIRS version 6 data sets were processed by and obtained from the Goddard Earth Services Data and Information Services Center (http://daac.gsfc.nasa.gov/). All rights reserved. Government sponsorship acknowledged.

10 *Competing interests.* The authors declare that they have no conflict of interest.

*Acknowledgements.* Part of this research was carried out at the Jet Propulsion Laboratory (JPL), California Institute of Technology, under a contract with the National Aeronautics and Space Administration. All authors were partially supported by the AIRS project at JPL.





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

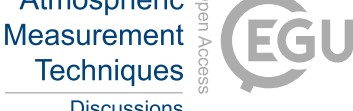

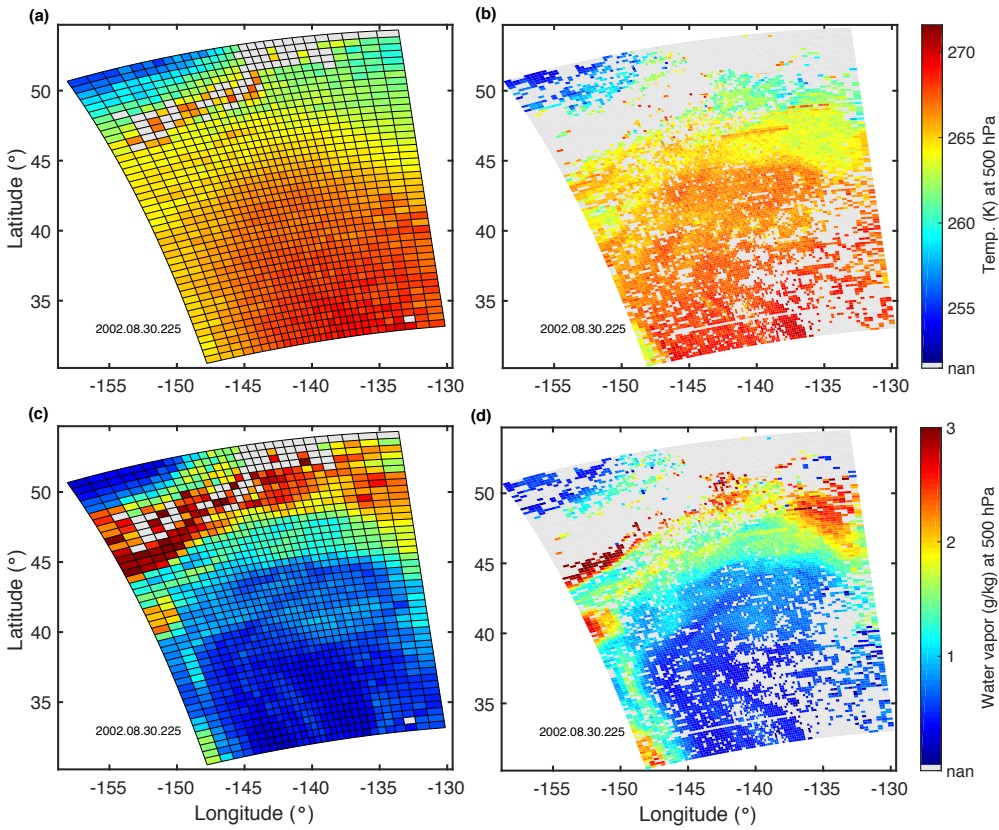

**Figure 1. (a)** Example of an AIRS-AMSU-HSBv6 temperature field at 500 hPa in a granule above the North Pacific Ocean - derived from soundings made during an ascending part of Aqua's orbit **(b)** the temperature field using AIRS-OE retrievals and **(c,d)** the corresponding moisture fields (the water vapor mass mixing ratio). Gray shading indicates that there was no acceptable retrieval.





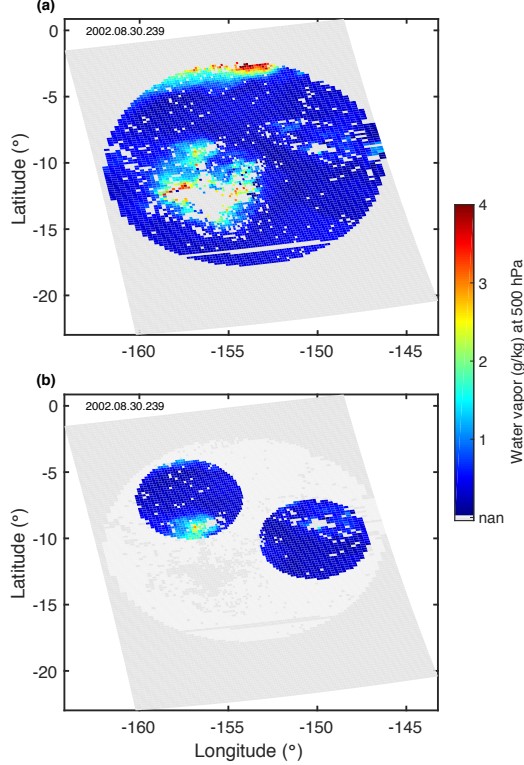

**Figure 2. (a)** Illustration of a water vapor mass mixing ratio field at 500 hPa using AIRS-OE retrievals that are inside a 15.4°-diameter circle and **(b)** inside two 6°-diameter circles. A dark gray shaded pixel inside a circle indicates that there was no acceptable water vapor content estimate. The granule, which is in an ascending part of Aqua's track above the South Pacific Ocean, has a relatively large yield (83%), however, it is clearly visible that atmospheric clouds are inhibiting retrievals around (156°W-13°S).





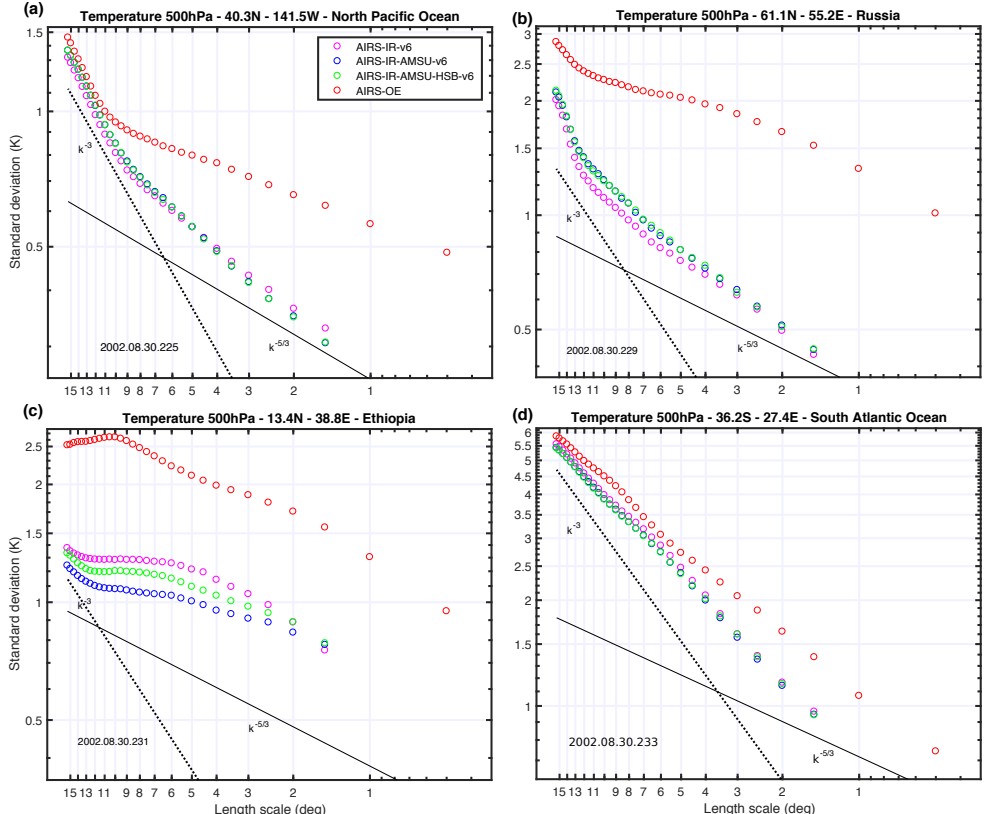

**Figure 3.** The standard deviation of the temperature at 500 hPa as a function of length scale $l$ using AIRS-IR (magenta open circles), AIRS-AMSU (blue open circles), AIRS-AMSU-HSB (green open circles) and AIRS-OE (red open circles) on double logarithmic axes with the $\beta = -5/3$ line (black solid line) and the $\beta = -3$ line (black dotted line) in a circular area with a diameter of 15.4° located **(a)** in the North Pacific Ocean **(b)** in Russia **(c)** in Ethiopia and **(d)** in the South Atlantic Ocean.





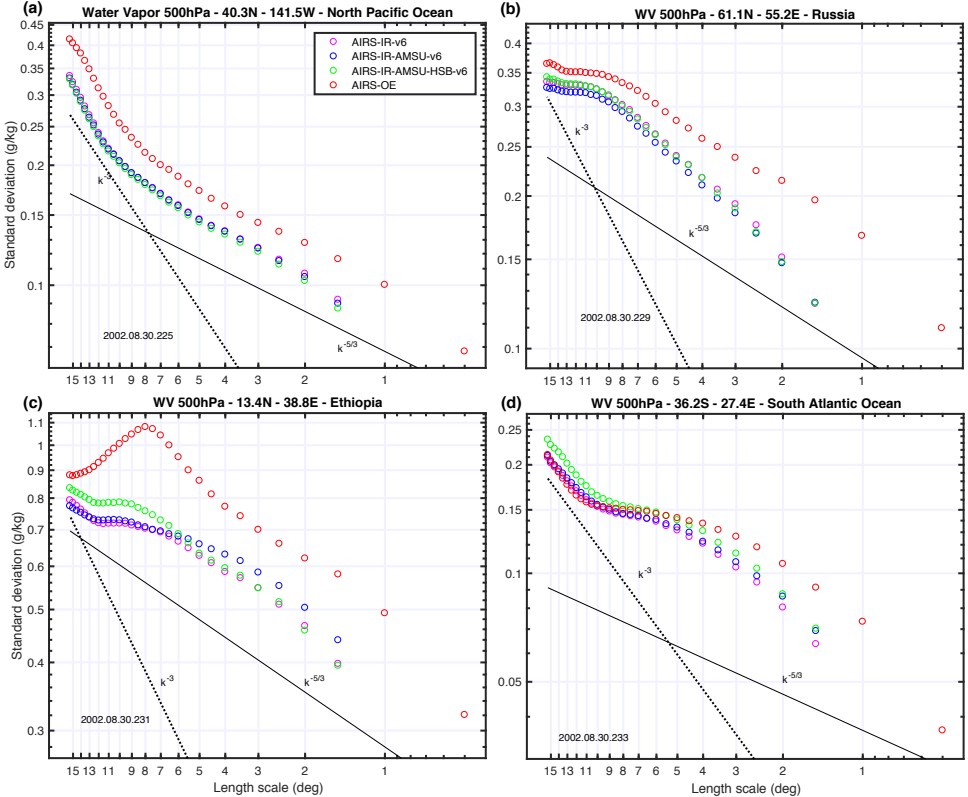

**Figure 4.** Same as Fig. 3 using the water vapor mass mixing ratio.





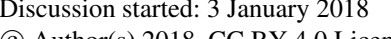

**Figure 5. (a)** AIRS-OE temperature 500 hPa variance scaling exponents $\alpha_L$, $\alpha_S$ and $\alpha_T$ as a function of Aqua's flight time. The right axes displays the $\beta$ value. **(b)** The standard deviations using length scales $l = 0.5°$ (lowest line) up to $L = 15.4°$ (usually the highest line) and **(c)** Aqua's longitude (left axis) and latitude (right axis).



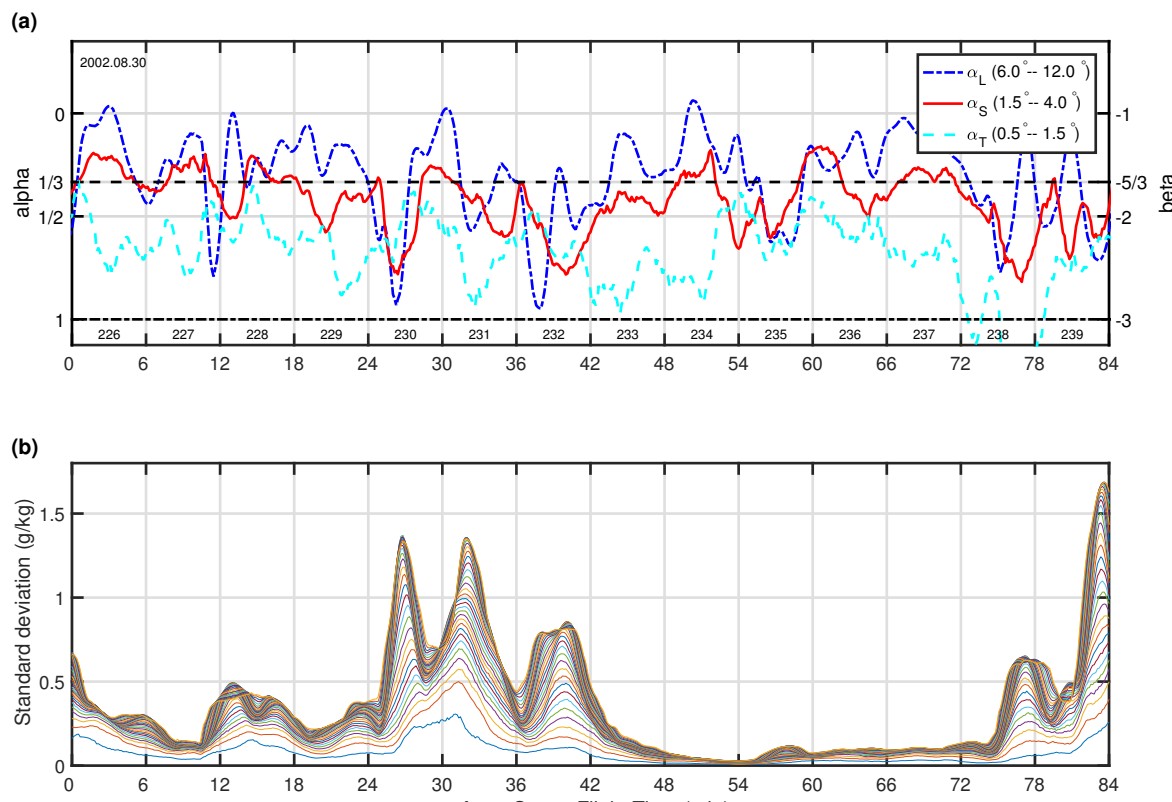

**Figure 6.** Same as Fig. 5 using the water vapor mass mixing ratio.





**Figure 7.** Variance scaling exponents $\alpha_L$ (blue lines), $\alpha_S$ (red lines) and $\alpha_T$ (cyan lines) as a function of Aqua's flight time for temperature **(a)** at 300 hPa **(b)** at 500 hPa and **(c)** at 850 hPa using AIRS-AMSU ($\alpha_L$ and $\alpha_S$) and AIRS-OE ($\alpha_T$). Additionally: the date (left upper corner) and granule numbers (bottom), the corresponding $\beta$ values can be seen on the right axis.





Figure 8. Same as Fig. 7 using the water vapor mass mixing ratio.





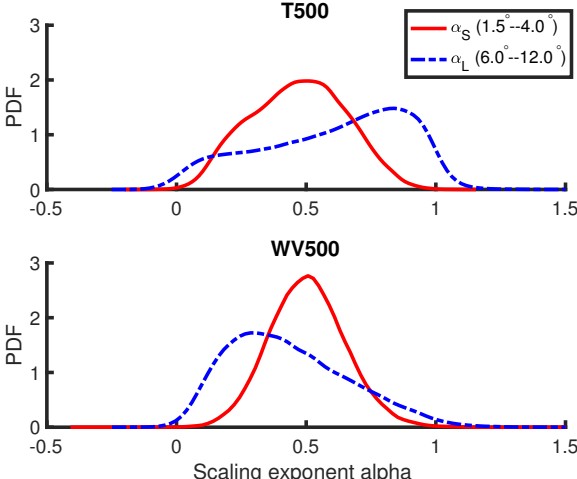

**Figure 9.** Probability density functions of AIRS-AMSUv6 (top panel) temperature 500 hPa and (bottom panel) water vapor 500 hPa scaling exponents $\alpha_L(6.0^\circ - 12.0^\circ)$ and $\alpha_S(1.5^\circ - 4.0^\circ)$ derived from five days of observations.



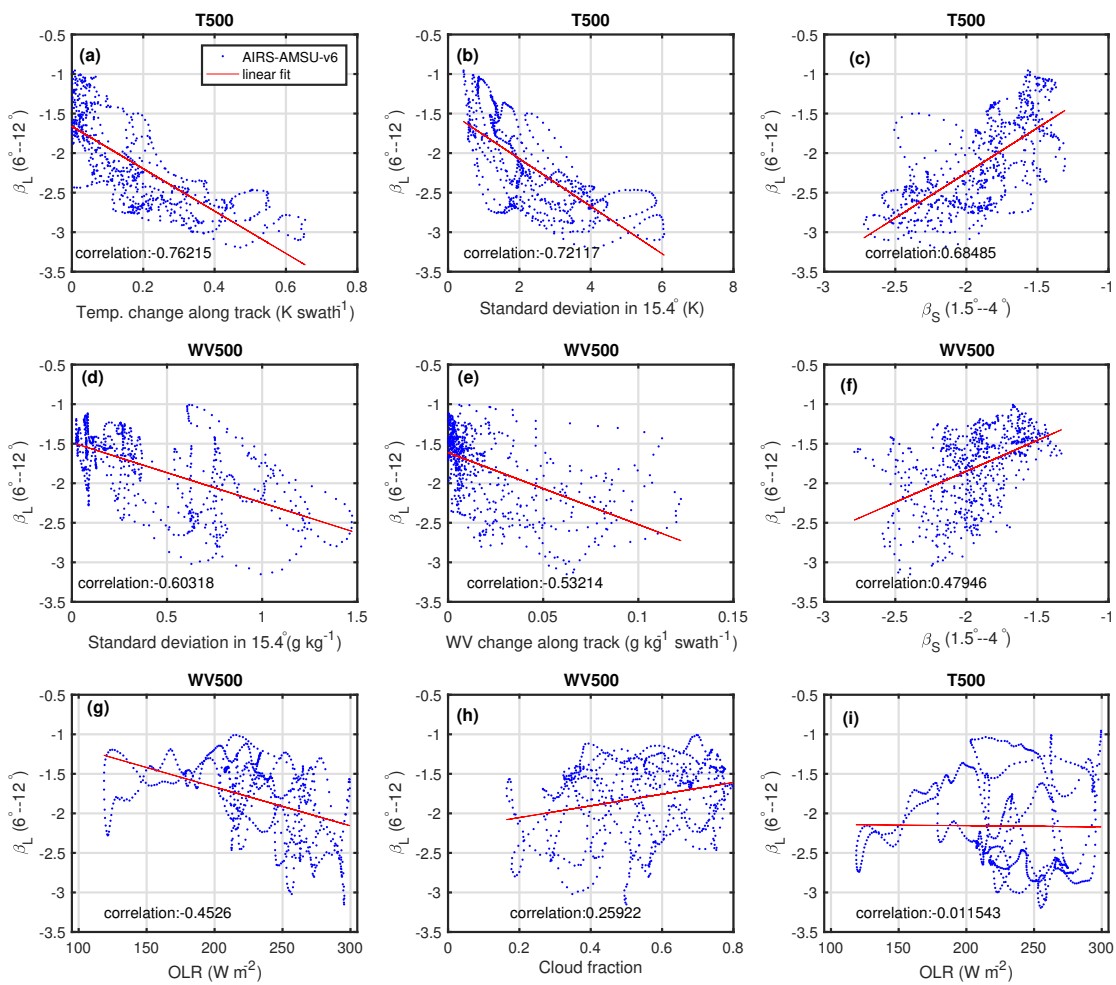

**Figure 10.** Scatterplots, linear fits and correlations between physical quantities and the slope $\beta_L$ derived using AIRS-AMSU temperatures at 500 hPa (T500) or water vapor at 500 hPa (WV500) along a 96 min segment of the Aqua track. Panels are ordered from strong **(a)** to weak **(i)** correlations.



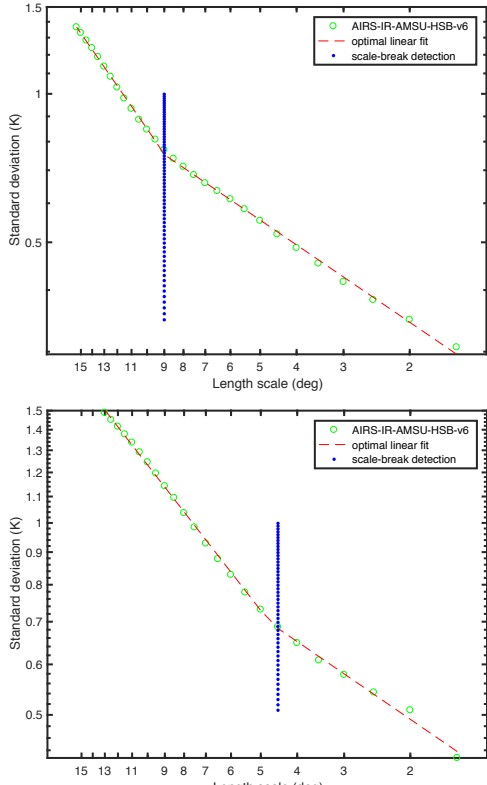

**Figure 11.** Two variance-scaling plots for AIRS-AMSU-HSB derived temperature at 500 hPa with detected scale breaks: (top panel) North Pacific Ocean (40.3N,141.5W) (bottom panel) North Pacific Ocean (44.6N,142.9W). In the top panel the slope changes at $9°$ and in the bottom panel at $4.5°$.







**Figure 12.** Probability density functions of the scale-break length scale using (left) temperature and (right) water vapor fields at three pressure levels.



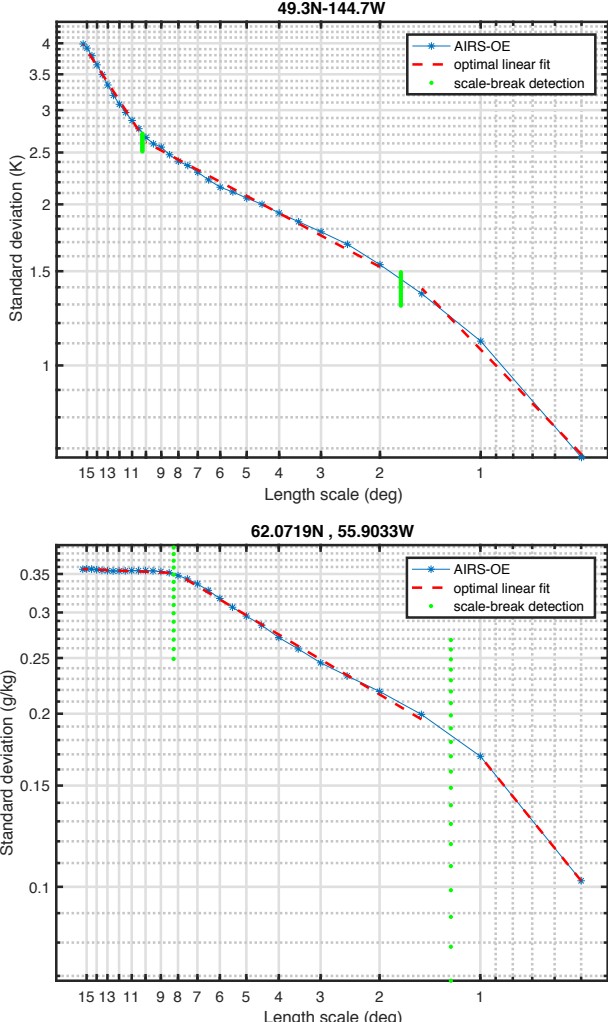

**Figure 13.** In these double-scale-break examples it can be seen that the slope can change at scales below $1.5°$. The standard deviations are calculated with AIRS-OE derived (top) temperature and (bottom) water vapor mass mixing ratio at 500 hPa.