# Peer review of "Instantaneous variance scaling of AIRS thermodynamic profiles using a circular area Monte Carlo approach"

_Atmospheric Measurement Techniques, 2017_

## Referee Comment (RC1) · Anonymous Referee #1 · 6 Feb 2018

GENERAL

The entire abstract will need to be rewritten in view of the results contained in the DOIs listed at the end of this review. It reflects omissions and mistaken assumptions listed in the detailed commentary that follows. The 'scaling of variance' has to recognise that the variance of atmospheric variables obtained by observations acquired by adequate measurement techniques does not converge. That is one of several relevant results obtained by the finding that atmospheric variables have non-Gaussian probability distributions, with fat-tailed power laws best represented by Lévy statistics. The zoo of scaling behaviours displayed in this manuscript arises from this basic fact and its consequences. These facts have to be at least recognised as being in existence rather than simply ignored as in the present manuscript.

DETAILED COMMENTARY

Page 1, Lines 16-18: Energy is deposited in the atmosphere by the absorption of photons by molecules, that is to say it has no alternative but to propagate upscale. This is argued at length in some of the references supplied as DOIs.

Lines 18 et seq: See the last, 8[th], DOI for a refutation of these arguments. They are profoundly mistaken.

Page 2, Lines 3-14: Inspection of the DOIs supplied will show a view differing substantially from that in these references. The Lindborg papers especially rest on bad assumptions. The Lovejoy & Schertzer book also has a lot on scaling in models, an advance on lines 15-28.

Section 3: KT09 in my opinion deploys flawed analytical methods. If the authors insist on using it, it must be justified in the light of the conclusions reached in the DOIs below. That includes the results on how easy it is to find false scale breaks, especially if less than three decades of good quality observations are present. They cannot be simply ignored.

Sections 4 and 5: At the very least these will have to be rewritten to accommodate the existence of alternative views and results conatined in the DOIs and books listed below. For example, models contain assumptions about variances and covariances being random that are at odds with observed reality; that is one of many problems.

REFERENCES

- https://doi.org/10.1080/01431161.2011.602652
- **DOI:** 10.1029/2012GL051689
- **DOI:** 10.1029/2009JD013353
- **DOI:** 10.1002/qj.644
- **DOI:** 10.1029/2008JD010651

- **DOI:** 10.1029/2007GL032122
- **DOI:** 10.1029/2007GL029359
- **DOI**: 10.5194/acp-12-327-2012
-

Finally, two books:

S Lovejoy & D Schertzer, 2013, THE WEATHER AND CLIMATE: Emergent Laws and Multifractal Cascades, CUP, ISBN 9781107018983

A F Tuck, 2008, ATMOSPHERIC TURBULENCE: A Molecular Dynamics Perspective, OUP, ISBN 9780199236534

---

## Referee Comment (RC2) · Anonymous Referee #2 · 6 Feb 2018

**Overall / general comments**

The paper presents a circular area Monte Carlo approach to assess scale invariance properties and scale breaks from AIRS measurements. Overall the paper is very well written but the statistics and correlations showed here are not always convincing. This is a promising technique but it needs to be apply to more data and to acknowledge the poor correlations observed in section 4.5 (and more data will help with correlations). Also, why sometime the authors use /alpha and sometime $\beta$? $\beta$ Is generally more known, especially when its concerns the well-known -5/3 value. There is no such reference for /alpha. I suggest using $\beta$ through the whole manuscript for consistency reasons.

Minor comments: Abstract Line 2: 13.5km is not really what I call "high spatial resolu-

tion". May be "higher" is better for the comparison with 45 km.

Introduction: Line 18: Please add also the reference: Kolmogorov, A. N.: "Dissipation of Energy in the Locally Isotropic Turbulence", Proceedings of the USSR Academy of Sciences (Russian), translated into English by Kolmogorov, Andrey Nikolaevich (8 July 1991), 23, 16–18, 1941.

2.2 Line 9: Why Retrieval System have their first letter in capital?

Figure 3: Please increase text/label font size It would be interesting to highlight (using arrow, line, marker, etc) the position of the scale break for each case. It would be more easy for the reader to see if there is a common off-set between the AIRS-xxx in the 4 locations.

Figure 4c: The large decreasing of standard deviation as a function of the length scale in the case AIRS-OE need to be more developed. This slope catches the eye directly when looking at the figure. This is probably due to small scale processes that are "resolved' with the higher resolutions but it should be mentioned.

4.5 Line 13: To me well-correlated is above 0.80, we can argue that the fig 10a is close to this value but then the correlation decrease. It becomes dangerous to me to talk about correlation below 0.7. This is especially true for water vapor where the values are too low. I can be simpler to remove WV from this plot and keep temperature only.

---

## Author Comment (AC1) · 20 Feb 2018

**1 Reply to reviewer #1**

*We are grateful for the helpful comments on our paper draft. Especially, the additional references (DOIs) improved our paper. Our point-by-point reply is in italic.*

Reviewer 1:

Review of *amt*2017463, Dorrestijn et al.

GENERAL

The entire abstract will need to be rewritten in view of the results contained in the DOIs listed at the end of this review. It reflects omissions and mistaken assumptions listed in the detailed commentary that follows. The scaling of variance has to recognise that the variance of atmospheric variables obtained by observations acquired by adequate measurement techniques does not converge. That is one of several relevant results obtained by the finding that atmospheric variables have non-Gaussian probability distributions, with fat-tailed power laws best represented by Lévy statistics. The zoo of scaling behaviours displayed in this manuscript arises from this basic fact and its consequences. These facts have to be at least recognised as being in existence rather than simply ignored as in the present manuscript.

*We choose to not rewrite the abstract, since it represents the findings of our research. The correlation between $\beta_L$ and the temperature slope along the track is -0.76, which is sufficiently strong to justify our assertions about the slopes. We remove the assertion that geostrophic turbulence is likely the cause of the $k^{-3}$ scaling.*

*About the convergence we will add to the manuscript (page 12, line 6) that: "another explanation for the large variety of exponents is the turbulent structure of the temperature and water vapor fields with long-tailed non-Gaussian distributions (Tuck 2010) that could inhibit correct estimation of the variance scaling exponents."*

DETAILED COMMENTARY

Page 1, Lines 16-18: Energy is deposited in the atmosphere by the absorption of photons by molecules, that is to say it has no alternative but to propagate upscale. This is argued at length in some of the references supplied as DOIs.

*We rewrote part of the introduction such that it is not necessary to include this comment*

Lines 18 et seq: See the last, 8th, DOI for a refutation of these arguments. They are profoundly mistaken. *We will omit this sentence to avoid any confusion.*

Page 2, Lines 3-14: Inspection of the DOIs supplied will show a view differing substantially from that in these references. The Lindborg papers especially rest on bad assumptions. The Lovejoy & Schertzer book also has a lot on scaling in models, an advance on lines 15-28.

*We mentioned the Lindborg paper because it reprints the Nastrom and Gage figure; We will add (page 2, line 13) the Lovejoy and Schertzer book to the list of numerical modeling and scaling paragraph.*

Section 3: KT09 in my opinion deploys flawed analytical methods. If the authors insist on using it, it must be justified in the light of the conclusions reached in the DOIs below. That includes the results on how easy it is to find false scale breaks, especially if less than three decades of good quality observations are present. They cannot be simply ignored.

*That KT09 works has been confirmed by Vogelzang 2015.*
*We add (page 12, line 21): "Lovejoy et al. (2009) and Pinel et al. (2012) showed that scale-breaks detected by aircrafts can be a result of anisotropy in the atmosphere; the transition to the -2.4 slope can indicate that at the larger scales the vertical scaling is detected instead of the horizontal. So, it can be the case that our scale-breaks are also a result of anisotropy."*

Sections 4 and 5: At the very least these will have to be rewritten to accommodate the existence of alternative views and results conatined in the DOIs and books listed below. For example, models contain assumptions about variances and covariances being random that are at odds with observed reality; that is one of many problems.

*We will add that there are alternative views and add the references in the discussion section.*

*For example:*

*In the discussion section we add (page 11, line 12): In Pinel et al. 2012, scale-breaks are found in the range 100 - 500 km with horizontal exponents that transition from -5/3 to -2.4. In the vertical direction they find the exponent -2.4, so they suggest that gently sloping isobaric aircraft trajectories cause the transition to -2.4.*

---

## Author Comment (AC2) · 20 Feb 2018

The comment was uploaded in the form of a supplement:
https://www.atmos-meas-tech-discuss.net/amt-2017-463/amt-2017-463-AC2-supplement.pdf
* * *

---

## Author Comment (AC3) · 28 Mar 2018

Overall / general comments

The paper presents a circular area Monte Carlo approach to assess scale invariance properties and scale breaks from AIRS measurements. Overall the paper is very well written but the statistics and correlations showed here are not always convincing. This is a promising technique but it needs to be apply to more data and to acknowledge the poor correlations observed in section 4.5 (and more data will help with correlations). Also, why sometime the authors use /alpha and sometime $\beta$? Is generally more known, especially when its concerns the well-known -5/3 value. There is no such reference for /alpha. I suggest using through the whole manuscript for consistency reasons.

*We agree that the datasets used for the correlation analysis are small. As this manuscript is meant to be a methodology rather than a full exploration of the potential of AIRS data, the computational expense of calculating the exponents is large, and the single infrared field of view retrieval of Irion et al. (2018) has not yet been operationalized, we are unable to go much further with more robust statistics in this initial study. In response to the reviewer comment, we have removed the three panels with the smallest correlation coefficients from Fig. 10. We argue that the six panels with the largest correlation coefficients are sufficiently large that, in our opinion, add value to the paper. We acknowledge that the usage of $\alpha$ and $\beta$ values could be somewhat confusing at times. Despite this, we choose to include them both in the new manuscript because many of the referenced studies use both types of exponents. The left and right axes on Figures 5–8 are intended to help guide the reader between these two exponents. The variance scaling exponents that we calculate are actually $\alpha$ values, also used by KT09, therefore we choose to show them. Omitting the $\beta$ values would devalue the paper, since these are better known.*

*We have added the following to the manuscript starting on Page 8 and line 33: "…the sample size from the limited set of granules is unable to yield a robust histogram. Our intent is to instead demonstrate the new scaling approach. A much larger and statistically robust dataset is outside the scope of this work."*

Minor comments: Abstract Line 2: 13.5km is not really what I call high spatial resolution. May be higher is better for the comparison with 45 km.

*We agree and use the word "higher".*

Introduction: Line 18: Please add also the reference: Kolmogorov, A. N.: Dissipation of Energy in the Locally Isotropic Turbulence, Proceedings of the USSR Academy of Sciences (Russian), translated into English by Kolmogorov, Andrey Nikolaevich (8 July 1991), 23, 16 18, 1941.

*Thanks. We now include this reference.*

2.2 Line 9: Why Retrieval System have their first letter in capital?

*This was a mistake. It should indeed be lowercase.*

Figure 3: Please increase text/label font size It would be interesting to highlight (using arrow, line, marker, etc) the position of the scale break for each case. It would be more easy for the reader to see if there is a common off-set between the AIRS-xxx in the 4 locations.

*We agree and changed the sizes. Scale breaks are introduced later in the paper and we think it would be too much information for the reader to digest if we put them into these figures before they are explained.*

Figure 4c: The large decreasing of standard deviation as a function of the length scale in the case AIRS-OE need to be more developed. This slope catches the eye directly when looking at the figure. This is probably due to small scale processes that are resolved with the higher resolutions but it should be mentioned.

*We agree this needs some additional description, and we added a comment to the manuscript starting on Page 7 and line 10: "In Fig. 4c, the discrepancies between the four retrievals are more significant at larger $l$, where AIRS-OE shows a decreasing standard deviation as a function of increasing $l$. However, the AIRS-OE with a peak around $8°$ may be a result of finer-scale fluctuations that are only captured by AIRS-OE."*

4.5 Line 13: To me well-correlated is above 0.80, we can argue that the fig 10a is close to this value but then the correlation decrease. It becomes dangerous to me to talk about correlation below 0.7. This is especially true for water vapor where the values are too low. I can be simpler to remove WV from this plot and keep temperature only.

*Per the earlier comments above, we removed the three panels with the lowest correlations from figure 10. We believe that the six panels shown in the figure adds additional value to the paper."*

---

## Author Comment (AC4) · 28 Mar 2018

**Reply to reviewer #1**

*We are grateful for the helpful comments on the submitted version of the manuscript. Our point-by-point replies are shown in italic font.*

*We have edited and rewritten much of the paper to improve the use of English and also to improve the clarity of the scientific content.*

Reviewer 1:
Review of *amt*2017463, Dorrestijn et al.

GENERAL
The entire abstract will need to be rewritten in view of the results contained in the DOIs listed at the end of this review. It reflects omissions and mistaken assumptions listed in the detailed commentary that follows. The scaling of variance has to recognise that the variance of atmospheric variables obtained by observations acquired by adequate measurement techniques does not converge. That is one of several relevant results obtained by the finding that atmospheric variables have non-Gaussian probability distributions, with fat-tailed power laws best represented by Lévy statistics. The zoo of scaling behaviours displayed in this manuscript arises from this basic fact and its consequences. These facts have to be at least recognised as being in existence rather than simply ignored as in the present manuscript.

*We thank the reviewer for this important perspective. After careful consideration of the material referenced in the DOIs, the authors fully agree with this assessment and furthermore have had an opportunity to identify appropriate modifications to the manuscript that we hope satisfactorily address the reviewer comments.*

*First, we acknowledge that the relevance of non-Gaussian distributions of temperature and specific humidity has indeed been addressed insufficiently in the submitted manuscript. We have added text and references to the manuscript and have added a statement reflecting this fact in the abstract. Second, we also acknowledge that the same holds true for the effects of 3D anisotropy and vertical scaling that can alias into horizontal variance scaling exponents on isobaric surfaces. The scale breaks may indeed be partly a result of the effects of vertical scaling. Therefore, we have rewritten the abstract and the entire manuscript and now reference the suggested DOI's.*

*Here is the new version of the abstract:*

*Satellite observations are used to obtain vertical profiles of variance scaling of temperature (T) and specific humidity (q) in the atmosphere. A higher spatial resolution retrieval at 13.5 km complements previous Atmospheric Infrared Sounder (AIRS) investigations with 45 km resolution retrievals and enables the derivation of power law scaling exponents to length scales as small as 55 km. We introduce a variable-sized circular-area Monte Carlo methodology to compute exponents instantaneously within the swath of AIRS that yields additional insight into scaling behavior. While this method is approximate and some biases are likely to exist within non-Gaussian portions of the satellite observational swaths of T and q, this method enables the estimation of scale-dependent behavior within instantaneous swaths for individual tropical and extratropical systems of interest. Scaling exponents are shown to fluctuate between $\beta = -1$ and -3 at scales $\geq$ 500 km, while at scales $\leq$ 500 km they are typically near $\beta \approx -2$, with q slightly lower than T at the smallest scales observed. In the extratropics, the large-scale $\beta$ is near -3. Within the tropics, however, the large-scale $\beta$ for T is closer to -1 as small-scale moist convective processes dominate. In the tropics, q exhibits large-scale $\beta$ between -2 and -3. The values of $\beta$ are generally consistent with previous works of either time-averaged spatial variance estimates, or aircraft*

*observations that require averaging over numerous flight observational segments. The instantaneous variance scaling methodology is relevant for cloud parameterization development and the assessment of time variability of scaling exponents.*

*The following modifications to the manuscript refer to the references in the list of DOIs:*

*Starting on Page 11, line 19: "The large variety of exponents is likely due to some extent from the turbulent structure of T and q fields with long-tailed non-Gaussian distributions (Tuck, 2010). These behaviors may inhibit the precise estimation of variance scaling exponents from observations. Further research is necessary to determine the impacts of the non-Gaussian distribution shapes of T and q on derived exponents, and their scale- dependence of non-Gaussianity. This effect may contribute to spreading out the PDFs of exponents."*

*Starting on Page 10, line 25: "Lovejoy et al. (2009) and Pinel et al. (2012) showed that scale-breaks detected by in situ aircraft observations may be the result of 3D anisotropy in atmospheric properties. In Pinel et al. (2012), scale breaks are observed in the 100 - 500 km range with horizontal exponents that transition from -5/3 to -2.4. In the vertical direction, an exponent of -2.4 is derived and suggests that gently sloping isobaric aircraft trajectories are the source of the transition to -2.4. Since the T and q exponents reside on isobaric surfaces (e.g. 500 hPa) in this work, one may expect that the vertical exponents may alias into the large-scale horizontal exponents. However, we do not find a clear indication of $\beta_L = -2.4$, although a focused effort on obtaining vertical scaling exponents with satellite soundings warrants further investigation. Unfortunately, the relative coarse vertical resolution of $\sim 2$ km from AIRS retrievals is not ideal for obtaining reliable estimates of vertical scaling exponents; dropsondes and radiosondes remain the standard and are much better suited to this observational challenge."*

DETAILED COMMENTARY

Page 1, Lines 16-18: Energy is deposited in the atmosphere by the absorption of photons by molecules, that is to say it has no alternative but to propagate upscale. This is argued at length in some of the references supplied as DOIs.

*We now refer to Tuck (2010) in which upscale energy propagation is discussed. We have added the following on Page 1, line 21: "a review paper on upscale energy propagation is found in Tuck (2010)."*

Lines 18 et seq: See the last, 8th, DOI for a refutation of these arguments. They are profoundly mistaken.

*We mention upscale energy transfer in 2D and quasi-geostrophic turbulence starting on page 1 line 18. In the 8th DOI (Schertzer et al. 2012) "Quasi-geostrophic turbulence and generalized scale invariance, a theoretical reply" give an alternative theory of energy transfer using fractal dimension turbulence. We add to the manuscript starting on page 1 and line 20: "Schertzer et al. (2012) give an alternative theory of energy transfer using fractal dimension turbulence."*

Page 2, Lines 3-14: Inspection of the DOIs supplied will show a view differing substantially from that in these references. The Lindborg papers especially rest on bad assumptions. The Lovejoy & Schertzer book also has a lot on scaling in models, an advance on lines 15-28.

*Although the reviewer suggests that the Lindborg paper may rest on bad assumptions, we mention the Lindborg paper because it reproduces the Nastrom and Gage figure to bolster its well-known results from aircraft data.*

Section 3: KT09 in my opinion deploys flawed analytical methods. If the authors insist on using it, it must be justified in the light of the conclusions reached in the DOIs below. That includes the results on how easy it is to find false scale breaks, especially if less than three decades of good quality observations are present. They cannot be simply ignored.

*First, we refer to the two additional revised blocks of text and references added above starting on Page 11 and line 19, and Page 10 and line 25.*

*Second, given the limitations stated above, the "poor man's spectral analysis" does have some advantages in particular for data sets as described in this manuscript that do not span three decades of scales. We address them in the short paragraph starting on Page 11 and line 11. "A major advantage of the "poor man's spectral analysis" method (Lorenz, 1979) is that relatively small datasets are sufficient to estimate variance scaling exponents. Reliable spectral power diagrams of observational data arise only after averaging over relatively large datasets. For instance, Nastrom and Gage (1985) obtained their spectral power diagrams by averaging over observations collected during 6,000 commercial aircraft flights. The calculation of spatial variances is still possible in the event of missing or poor quality data, in which case conventional spectral analysis cannot be employed (Vogelzang et al., 2015)"*

*Third, the reviewer is right in that we need clearer justification for using this methodology. In the abstract we have now made clearer that we need instantaneous estimates of the scale-dependent variance: "... this method enables the estimation of scale-dependent behavior within instantaneous swaths for individual tropical and extratropical systems of interest."*

*Furthermore, we have enhanced the discussion at the end of the paper on why the authors think this approach is meritorious, despite its limitations and need for further investigation. At this time, we are not aware of any other approaches that can be used to exploit satellite soundings of T and q to quantify individual storm evolution and structure of scaling exponents, which are inherently tied to their predictability. We have significantly added to and revised the last paragraph starting on Page 12 and line 5:*
*"This novel instantaneous variance scaling methodology may enable detailed examination of the variance scaling of the time evolution of storm systems, such as extratropical cyclones at different stages in their lifecycle as previously demonstrated with numerical simulations by Waite and Snyder (2013), or with deep convection along the Mei-Yu front by Peng et al. (2014). The changes in the kinetic energy spectra in Waite and Snyder (2013) and Peng et al. (2014) occur on time scales of hours to several days. We postulate that scaling exponents derived from instantaneous snapshots obtained from satellite swath data will be useful observational constraints for time-dependent spectra generated from numerical modeling experiments. To conclude, it is well known that the time scale of predictability is closely linked to the spatial scale of the phenomenon of interest (Lorenz, 1969). In the case of moist baroclinic waves, steeper (shallower) spectral slopes at small scales for individual baroclinic waves are inherently more (less) predictable as the slope portrays the relative importance of convection within any given disturbance (Zhang et al., 2007). As a result, the instantaneous scaling exponents are expected to potentially offer a new type of observational constraint with relevance to the predictability of individual tropical or extratropical disturbances."*

Sections 4 and 5: At the very least these will have to be rewritten to accommodate the existence of alternative

views and results conatined in the DOIs and books listed below. For example, models contain assumptions about variances and covariances being random that are at odds with observed reality; that is one of many problems.

*We hope that the revisions and responses described above generally address these concerns. We have rewritten much of the paper to improve clarity of content and have added and revised many paragraphs taking into account the suggested DOIs (which were very helpful). The reviewer comment above regarding models (we assume numerical) is well taken. There are numerous issues of interest in numerical models, but those aspects are well beyond the scope of this article. We hope to advance this methodology on evaluating numerical model output in the near future, however.*